# The effectiveness of public health interventions against COVID-19: Lessons from the Singapore experience

John P. Ansah[1,2]*, David Bruce Matchar[1,3,4], Sean Lam Shao Wei[1,5], Jenny G. Low[6,7], Ahmad Reza Pourghaderi[1,5], Fahad Javaid Siddiqui[1], Tessa Lui Shi Min[1], Aloysius Chia Wei-Yan[1], Marcus Eng Hock Ong[1,8]

1 Programme in Health Services and Systems Research, Duke-NUS Medical School, Singapore, Singapore,
2 Residential College 4, National University of Singapore, Singapore, Singapore, 3 Department of Internal Medicine (General Internal Medicine), Duke University, Singapore, Singapore, 4 Department of Internal Medicine, Singapore General Hospital, Singapore, Singapore, 5 Health Services Research Centre, Singapore Health Services, Singapore, Singapore, 6 Department of Infectious Diseases, Singapore General Hospital, Singapore, Singapore, 7 Programme in Emerging Infectious Diseases, Duke-NUS Medical School, Singapore, Singapore, 8 Department of Emergency Medicine, Singapore General Hospital, Singapore, Singapore

* john.ansah@duke-nus.edu.sg

**Data Availability Statement:** ALL DATA FILES USED IN THIS PAPER ARE PUBLICLY AVAILABLE AND THE SOURCES ARE CITED IN THE PAPER AND SUPPORTING INFORMATION FILES.

## Abstract

### Background

In dealing with community spread of COVID-19, two active interventions have been attempted or advocated—containment, and mitigation. Given the extensive impact of COVID-19 globally, there is international interest to learn from best practices that have been shown to work in controlling community spread to inform future outbreaks. This study explores the trajectory of COVID-19 infection in Singapore had the government intervention not focused on containment, but rather on mitigation. In addition, we estimate the actual COVID-19 infection cases in Singapore, given that confirmed cases are publicly available.

### Methods and findings

We developed a COVID-19 infection model, which is a modified SIR model that differentiate between detected (diagnosed) and undetected (undiagnosed) individuals and segments total population into seven health states: susceptible (S), infected asymptomatic undiagnosed (A), infected asymptomatic diagnosed (I), infected symptomatic undiagnosed (U), infected symptomatic diagnosed (E), recovered (R), and dead (D). To account for the infection stages of the asymptomatic and symptomatic infected individuals, the asymptomatic infected individuals were further disaggregated into three infection stages: (a) latent (b) infectious and (c) non-infectious; while the symptomatic infected were disaggregated into two stages: (a) infectious and (b) non-infectious. The simulation result shows that by the end of the current epidemic cycle without considering the possibility of a second wave, under the containment intervention implemented in Singapore, the confirmed number of Singaporeans infected with COVID-19 (diagnosed asymptomatic and symptomatic cases) is projected to

**Funding:** The production of this manuscript was funded by the Singapore Ministry of Health's National Medical Research Council.

**Competing interests:** The authors have declared that no competing interests exist.

be 52,053 (with 95% confidence range of 49,370–54,735) representing 0.87% (0.83%-0.92%) of the total population; while the actual number of Singaporeans infected with COVID-19 (diagnosed and undiagnosed asymptomatic and symptomatic infected cases) is projected to be 86,041 (81,097–90,986), which is 1.65 times the confirmed cases and represents 1.45% (1.36%-1.53%) of the total population. A peak in infected cases is projected to have occurred on around day 125 (27/05/2020) for the confirmed infected cases and around day 115 (17/05/2020) for the actual infected cases. The number of deaths is estimated to be 37 (34–39) among those infected with COVID-19 by the end of the epidemic cycle; consequently, the perceived case fatality rate is projected to be 0.07%, while the actual case fatality rate is estimated to be 0.043%. Importantly, our simulation model results suggest that there about 65% more COVID-19 infection cases in Singapore that have not been captured in the official reported numbers which could be uncovered via a serological study. Compared to the containment intervention, a mitigation intervention would have resulted in early peak infection, and increase both the cumulative confirmed and actual infection cases and deaths.

## Conclusion

Early public health measures in the context of targeted, aggressive containment including swift and effective contact tracing and quarantine, was likely responsible for suppressing the number of COVID-19 infections in Singapore.

## Introduction

In late December 2019, the coronavirus disease 2019 (COVID-19) was reported in Wuhan, China. This became the first epicenter of COVID-19, leading to a lockdown of Wuhan after human-to-human transmission was confirmed. The rapid increase in the number of infected persons in China and globally thereafter led the World Health Organization (WHO) to declare a public health emergency of international concern on January 30, 2020, and a pandemic on March 11, 2020, as it became increasingly evident that COVID-19 had spread globally [1].

According to the World Health Organization, as of June 11, 2020, 7,273,958 confirmed COVID-19 cases and 413,372 deaths have been reported globally [2]. The United States (USA) as of June 11, 2020, reports the highest number of confirmed cases of 1,968,331, and deaths (111,978). Human-to-human transmission will be difficult to suppress as infected individuals may be able to transmit the virus days before experiencing significant symptoms.

In dealing with community spread of COVID-19, two active interventions have been attempted or advocated. The first is "containment", involving quarantine of specific individuals based on tracing from their contact to a known infected individual or their history of recent travel to a high prevalence country or region. "Mitigation" is a second strategy aiming to limit movement at the population level; social distancing ranges from limiting physical proximity between people to no less than one meter to community lockdown. At different points in the progression of COVID-19, many countries have implemented various policy strategies, with most applying a mixture of containment and mitigation to reduce disease burden, morbidity and mortality when faced with local exponential growth of infected cases, all whilst aiming to minimize social and economic disruption. An additional major consideration in policy discussions is how these interventions mitigates stress on healthcare systems so that essential medical care can be provided to non-COVID as well as COVID patients. This is the rationale for

pursuing interventions that might not substantially reduce total numbers of infections but would rather "flatten the curve".

The city-state of Singapore was one of the first countries to record a confirmed case of COVID-19 shortly after the outbreak in China. In response, the Singaporean government adopted an aggressive containment strategy focusing mainly on swift and effective contact tracing and quarantine of individuals in order to prevent small clusters of COVID-19 infection from amplifying in a chainlike fashion into widespread community transmission. The containment strategy implemented by Singapore has been associated with a more moderate rise in number of infections than otherwise expected; the moderate number has allowed the Singapore health system to meet the needs of the confirmed COVID-19 cases, resulting in very few (25) deaths as of June 7, 2020. Given the extensive impact of COVID-19 globally, there is international interest to learn from best practices that have shown to work in controlling community spread to inform future epidemic outbreaks. This research aims to explore what the trajectory of COVID-19 infection would have been in Singapore had the government intervention not focused on containment, but rather on mitigation to inform future epidemic outbreak interventions.

## Singapore's approach to COVID-19

Guided by the experience of Severe Acute Respiratory Syndrome (SARS) in 2003 which led to 228 cases and 33 deaths in Singapore, the city-state had since ramped up its infectious disease prevention measures in combating future epidemics [3, 4]. The relative success behind Singapore's containment of the pandemic, leading to only 25 deaths thus far, can be attributed to the government's quick response, immediate contact tracing, targeted quarantine measures, strict quarantine management, and abundant community communication. Following the identification of the outbreak in Wuhan, China, the Singaporean government acted immediately to ensure the safety of their citizens [5]. Fig 1 shows the timeline of actions taken in Singapore in response to the COVID-19 outbreak. Having learned from the experience of combating SARS, the Singaporean government developed physical and operational infrastructure to support rapid contact tracing, quarantine, and medical services for infected individuals; the revision of the Infectious Disease Act (IDA) ensures that all measures needed to control any future outbreaks could be implemented [3].

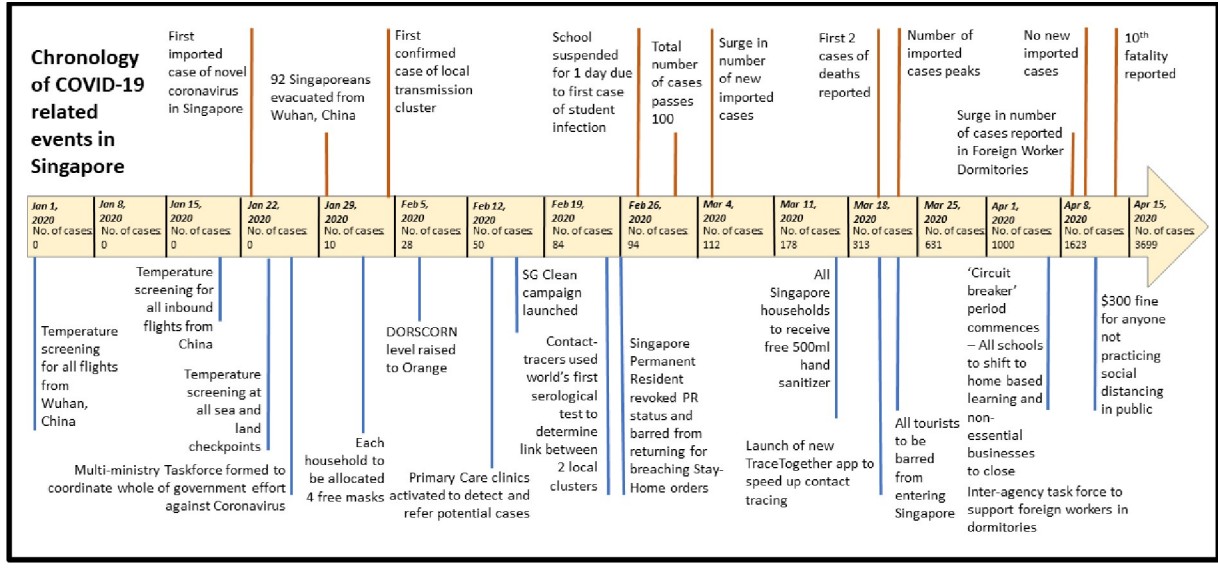

**Fig 1. Timeline of coronavirus actions taken in Singapore.**

The initial success in containing the spread of COVID-19 has been attributed to the efficient and immediate contact tracing of patients who had been diagnosed with the virus [6]. Although contact tracing is by no means a new innovation, Singapore's aggressive and proactive approach, praised by the WHO, has led to the avoidance of a community-wide spread [7]. Upon receiving word of a newly diagnosed patient, contact tracers immediately embark on a labor-intensive attempt to identify people who have been in contact with infected individuals, thereby being able to find those who may themselves may be infected [6]. Because the incubation period of COVID-19 is relatively short, contact tracing must be swift in order to contain the outbreak. Security cameras, receipts, and work calendars are used to fill in the gaps in memory of those infected who are unable to recall their whereabouts. Launched on March 20, the Trace Together application was developed to facilitate contact tracing by monitoring users' locations and alerting any user who has come into contact with any individual who has tested positive for COVID-19. A detailed description of Singapore's approach to COVID-19 and COVID-19 cases is provided in the online only S1 and S2 Appendices in S1 File.

## Methods

Several simulation models have been developed to predict the infection trajectory of the COVID-19 pandemic. Lin and colleagues developed a modified SEIR (susceptible, exposed, infectious, removed) model considering risk perception [8], while Casella developed a control-oriented SIR model that stresses the effects of delays [3], and Wu and colleagues used transmission dynamics to estimate the clinical severity of COVID [9]. Giordano and colleagues developed a simulation model referred to as SIDARTHE that disaggregated the total population into eight stages to explore the impact of population-wide interventions [10]. Stochastic transmission models have also been considered [11]. Flexman and colleagues [12] used the discrete renewal process approach—which is related to the SIR model, an approach that has a strong theoretical basis in stochastic individual-based counting process—to model the number of infections over time [13].

We developed a COVID-19 infection model, which is a modified SIR model [13] similar to the SIDARTHE model [10], that differentiate between detected (diagnosed) and undetected (undiagnosed) individuals and segments total population into seven health states: susceptible (S), infected asymptomatic undiagnosed (A), infected asymptomatic diagnosed (I), infected symptomatic undiagnosed (U), infected symptomatic diagnosed (E), recovered (R), and dead (D). To account for the infection stages of the asymptomatic and symptomatic infected individuals, the asymptomatic infected individuals were further disaggregated into three infection stages: (a) latent—the infection stage before becoming infectious; (b) infectious; and (c) non-infectious—when the virus is no longer viable [14]. Likewise, the symptomatic infected individuals were disaggregated into two infection stages: (a) infectious and (b) non-infectious. The main differences between the proposed modified SIR model presented herein and the SIDARTHE model are that: (a) our model has 7 health states compared to the eight-health state of the SIDARTHE model. (b) our modified SIR model further disaggregates the asymptomatic infected individuals into three infection stages, which is not the case for SIDARTHE model. (c) the modified SIR model disaggregates the symptomatic infected individuals into two infection stages, which is absent in the SIDARTHE model. We strongly believe that the amendment made compared to the SIDARTHE model will significantly improve the accuracy of the simulation model to predict the infection trajectory of COVID-19, and more importantly, help to estimate the actual COVID-19 infection cases. Briefly, the transmission dynamics as depicted in Figs 2 and 3 is as follows: through exposure to an infected individual, some of the exposed individuals become infected, and transition to the infected asymptomatic health state. At the asymptomatic health state, infected individuals go through three infection stages—latent, infectious

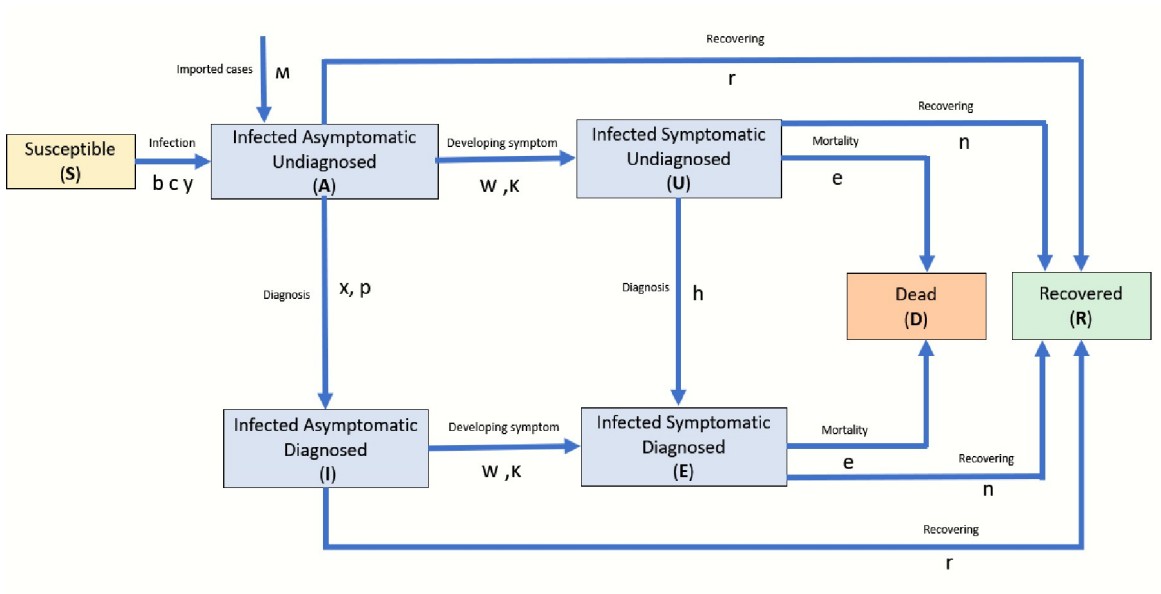

**Fig 2. The simulation model structure representing the dynamic interactions of different health state of COVID-19.**

and non-infectious [14]. All newly infected individuals move from asymptomatic latent state to asymptomatic infectious stage. During the asymptomatic infectious stage, some individuals will develop symptoms and transition from asymptomatic to symptomatic infectious state, while others will transition to non-infectious asymptomatic and eventually recover from the virus. Infected individuals in asymptomatic and symptomatic health state can be diagnosed through testing and move from undiagnosed to diagnosed health state. Non-infectious infected individuals (asymptomatic and symptomatic) will overtime recover and move to the recovered health state; infectious symptomatic infected individuals may either recover or die from the infection.

The immune response including duration of immunity for COVID-19 infection is not yet understood [15]. Duration of antibody responses against other human coronaviruses may be

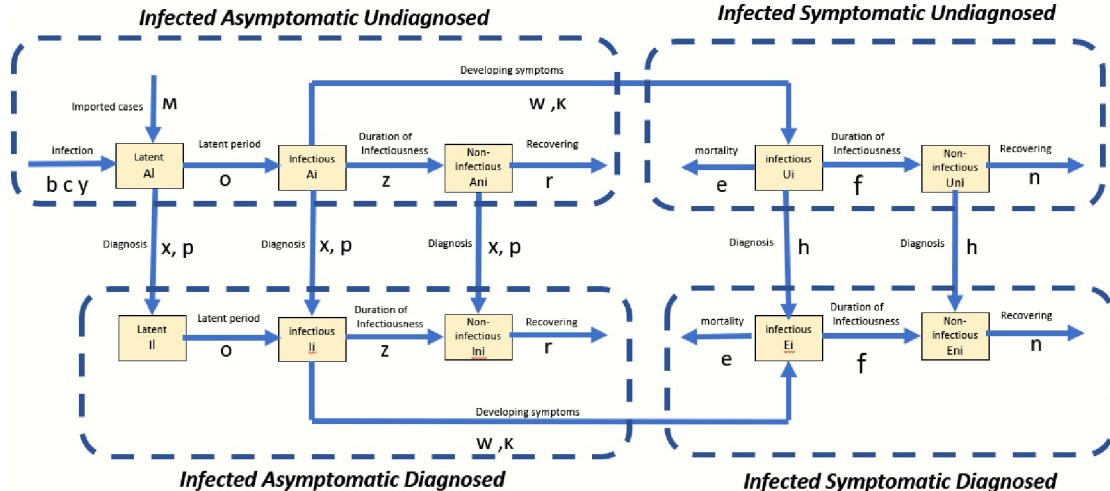

**Fig 3. Detailed model structure for asymptomatic and symptomatic undiagnosed and diagnosed infected health state and their interactions.**

relevant in this context. For example, following infection with SARS-CoV-1 (the virus that caused SARS), concentrations of neutralizing antibodies (NAbs) remained high for approximately 4 to 5 months before subsequently declining slowly during the next 2 to 3 years [16]; while neutralizing antibodies (NAbs) following infection with MERS-CoV (the virus that caused Middle East respiratory syndrome) have persisted up to 34 months in recovered patients [17]. But it is not yet known whether similar immune protection will be observed for individuals infected with COVID-19. For the purpose of this study, we are interested in projections over a relatively short time horizon (365 days) within which temporary immunity is likely still to be in place. As a result, we assumed a zero probability of becoming susceptible again after recovering from the infection over the simulation time, although anecdotal evidence of re-infection is found in literature [18]. In addition, our simulation model specifically makes a distinction between diagnosed and undiagnosed infected individuals. Our model, assume that only undiagnosed infectious individuals create infection and that due to proper isolation and compliance to strict rules diagnosed infectious individuals do not create infection.

The COVID-19 epidemic in Singapore can be separated into two separate outbreaks: one in the general community (herein referred to as local) and the other among the foreign workers living in dormitories (referred to as migrants). Hence, two simulation models with the same structure (described in Figs 2 and 3) but with slightly different model inputs are used, as indicated in the model input table. The distinction between the local and migrant workers is important to capture the infection dynamics in these two populations and the different policies implemented to control the outbreaks. The ordinary differential equations that describe the evolution of the population in each health state over time are described below:

$$\frac{dS}{dt} = -c\,S(t)\,y\left(\frac{Ai(t) + Ui(t)}{T(t)}\right) \tag{1}$$

$$\frac{dAl}{dt} = c\,S(t)\,y\left(\frac{Ai(t) + Ui(t)}{T(t)}\right) + \text{м}(t) - \left(\frac{p}{x}\right)Al(t) - \left(\frac{1}{o}\right)Al(t) \tag{2}$$

$$\frac{dAi}{dt} = \left(\frac{1}{o}\right)Al(t) - \left(\frac{1}{w}\right)\left[(1 - \text{к})Al(t) - \left(\frac{p}{x}\right)Ai(t)\right] - \left(\frac{p}{x}\right)Al(t) - \left(\frac{1}{z}\right)Ai(t) \tag{3}$$

$$\frac{dAni}{dt} = \left(\frac{1}{z}\right)Ai(t) - \left(\frac{p}{x}\right)Ani(t) - \left(\frac{1}{r}\right)Ani(t) \tag{4}$$

$$\frac{dIl}{dt} = \left(\frac{p}{x}\right)Al(t) - \left(\frac{1}{o}\right)Il(t) \tag{5}$$

$$\frac{dIi}{dt} = \left(\frac{1}{o}\right)Il(t) + \left(\frac{p}{x}\right)Ai(t) - \left(\frac{1}{z}\right)Ii(t) - \left(\frac{1}{w}\right)\left[(1 - \text{к})Il(t) - \left(\frac{p}{x}\right)Ii(t)\right] \tag{6}$$

$$\frac{dIni}{dt} = \left(\frac{1}{z}\right)Ii(t) + \left(\frac{p}{x}\right)Ani(t) - \left(\frac{1}{r}\right)Ini(t) \tag{7}$$

$$\frac{dUi}{dt} = \left(\frac{1}{w}\right)\left[(1 - \text{к})Al(t) - \left(\frac{p}{x}\right)Ai(t)\right] - \left(\frac{1}{f}\right)Ui(t) - \left(\frac{1}{h}\right)Ui(t) - e\text{Ui}(t) \tag{8}$$

$$\frac{dUni}{dt} = \left(\frac{1}{f}\right)Ui(t) - \left(\frac{1}{n}\right)Uni(t) - \left(\frac{1}{h}\right)Uni(t) \tag{9}$$

$$\frac{dEi}{dt} = \left(\frac{1}{w}\right)\left[(1-\kappa)Il(t) - \left(\frac{p}{x}\right)Ii(t)\right] + \left(\frac{1}{h}\right)Ui(t) - \left(\frac{1}{f}\right)Ei(t) - eEi(t) \tag{10}$$

$$\frac{dEni}{dt} = \left(\frac{1}{f}\right)Ei(t) + \left(\frac{1}{h}\right)Ui(t) - \left(\frac{1}{n}\right)Eni(t) \tag{11}$$

$$\frac{dR}{dt} = \left(\frac{1}{r}\right)Ani(t) + \left(\frac{1}{r}\right)Ini(t) + \left(\frac{1}{n}\right)Uni(t) + \left(\frac{1}{n}\right)Eni(t) \tag{12}$$

$$\frac{dD}{dt} = eUi(t) + eEi(t) \tag{13}$$

The parameters are defined as follows:

- $b$ denote the proportion of undiagnosed infected individuals (asymptomatic and symptomatic) in the infectious stage. We assumed herein that diagnosed infected individuals (asymptomatic and symptomatic) do not create infection because in Singapore, all diagnosed infected COVID-19 patients are properly isolated or quarantined in hospitals or community isolation centers and comply to strict rules until they are determined recovered. $c$ is contact frequency of susceptible individuals and $y$ is the probability of infection given a contact with infected individual. Thus, $b\,c\,y$ indicates the probability of disease transmission given contact with infected individuals. Contact frequency was estimated to be 8 close contacts per person per day at the start of the infection and reduced to 4 close contact per person per day during the nationwide lockdown (known in Singapore as Circuit Breaker) for the local community; contact frequency for the migrant workers were estimated to be 10 close contact per person per day. The contact frequency of migrant workers remained unchanged during the lockdown period due to their living arrangements.

- $o$, $z$, x, $p$, $w$, M, $\kappa$, and r, respectively, denote latent period (the time it takes to become infectious), duration of infectiousness (asymptomatic), onset to isolation delay, fraction quarantined, incubation time to develop symptoms, imported cases, fraction asymptomatic not developing symptoms and average recovery time (undiagnosed and diagnosed asymptomatic). Duration of infectiousness for asymptomatic infected individuals is shorter compared to symptomatic infected individuals. The rate of diagnosis via testing for asymptomatic infected individuals are expected to be lower compared to symptomatic infected individuals because the probability of testing individuals with symptoms is much higher compared to those without symptoms. Average recovery time is a reflection of criteria for discharge and denotes the time non-infectious infected individuals are declared recovered.

- $f$,$h$,$e$, and $n$, denote respectively duration of infectiousness (symptomatic), symptoms development to care delay (symptomatic), mortality rate (undiagnosed and diagnosed symptomatic), and average recovery time (undiagnosed and diagnosed symptomatic). Mortality rate for undiagnosed and diagnosed symptomatic infected individuals, were assumed to be the same. The model assumes higher probability of diagnosis among symptomatic infected individuals compared to asymptomatic infected individuals.

- *Al*, *Ai*, *and Ani* denote infected asymptomatic undiagnosed latent, infected asymptomatic undiagnosed infectious and infected asymptomatic undiagnosed not-infectious.

- *Il*, *Ii*, *and Ini* denote infected asymptomatic diagnosed latent, infected asymptomatic diagnosed infectious and infected asymptomatic diagnosed not-infectious.

- *Ui*, *and Uni* demote infected symptomatic undiagnosed infectious and infected symptomatic undiagnosed not-infectious.

- *Ei*, *and Eni* demote infected symptomatic diagnosed infectious and infected symptomatic diagnosed not-infectious.

## Model inputs

Table 1 shows the input parameters for the COVID-19 simulation model for the reference case, and sources of the input parameters. COVID-19 Singapore data from 23rd January to 7th June 2020 was used to estimate some of the model inputs including contact frequency, mortality rate, and imported cases. COVID-19 data used was fully anonymized before it was accessed. In addition, published evidence from Singapore and other countries were used for other parameters as shown in Table 1. The endogenously estimated reproduction number reflects the progressive introduction of policies to control the infection. At the start of the infection the reproduction number for the local infection was $R_0 = 2$, while that for the migrant workers was $R_0 = 3$, which resulted in increased number of cases. After the introduction of nationwide lockdown (circuit breaker) to enforce social distancing, enhanced contact tracing and isolation and compulsory wearing of face-mask, in addition to increased hygiene and behavioural change awareness, the estimated reproduction number for the local infection decreased to $R_0 = 0.73$ while that for the migrant workers was $R_0 = 1.5$. The endogenous parameter "proportion of undiagnosed infected" is defined as the sum of "infected asymptomatic undiagnosed infectious" and "infected symptomatic undiagnosed infectious", divided by "total population.

**Table 1. Model inputs (parameters with * were included in the sensitivity analysis and varied ±25%).**

| Parameter | | Values (local) | Values (migrant) | Reference |
|---|---|---|---|---|
| Proportion of undiagnosed infected | b | endogenous | endogenous | |
| Contact frequency per person | c | 8–2 persons | 10 persons | Ministry of Health, Singapore (2020) [19] |
| Probability of infection given contact* | y | 0.039 | 0.039 | Lei Luo (2020) [20] |
| Latent period * | o | 2.3 days | 2.3 days | Xi He et al, 2020 [21] |
| Duration of infectiousness asymptomatic* | z | 4.5 days | 4.5 days | Calibrated |
| Onset to isolation delay* | x | 9–2 days | 9–2 days | Ng, Y. et al (2020) [22] |
| Incubation time* | w | 5 days | 5 days | Pung et al (2020) [23] |
| Recovery time diagnosed asymptomatic* | r | 10 days | 10 days | Calibrated |
| Recovery time undiagnosed asymptomatic* | r | 10 days | 10 days | Calibrated |
| Duration of infectiousness symptomatic* | *f* | 13 days | 13 days | National Centre for Infectious Diseases (2020) [24] |
| Symptoms development to care delay* | h | 5.5–2 days | 5.5–2 days | Steven Sanche et al (2020) [25] |
| Mortality rate undiagnosed symptomatic* | e | 0.004 | 0.004 | Ministry of Health, Singapore (2020) [19] |
| Mortality rate diagnosed symptomatic* | e | 0.004 | 0.004 | Ministry of Health, Singapore (2020) [19] |
| Recovery time undiagnosed symptomatic* | *n* | 10 days | 10 days | Calibrated |
| Recovery time diagnosed symptomatic* | *n* | 10 days | 10 days | Calibrated |
| Fraction asymptomatic without symptoms* | κ | 0.7 | 0.7 | Michael Day (2020) [26] |
| Imported cases | M | Time series | 0 | Ministry of Health Singapore (2020) [19] |
| Fraction quarantined | p | endogenous | endogenous | |

Likewise, the endogenous parameter "fraction quarantined" is defined as "total confirmed cases" divided by "total infected cases".

## Simulated interventions

As a reference case, we simulated the spread of COVID-19 based on extant Singapore policy, denoted "Singapore containment intervention". This was then compared to two counterfactual interventions to estimate the impact Singapore might have experienced under alternative interventions, denoted "mitigation: physical distancing with low quarantine rate"; and "mitigation: physical distancing with moderate quarantine rate". For details of COVID-19 mitigation and containment interventions see the literature as cited [27, 28].

**Singapore containment approach.** As noted, the interventions implemented in Singapore to manage COVID-19 focused mainly on containment, emphasizing a swift and meticulous approach to contact tracing and isolate. In the model, onset to isolation delay decreased from 9 days early in the outbreak to 2 days as provided in the literature as cited [21]. In addition, social distancing interventions put in place that discourage large group gathering, separation in public, and the promotion of telecommuting decreased the number of close contacts per person from 8 to 4 for the local community and that for the migrant workers remain at 10. Due to the living arrangement of the migrant workers, confirmed cases were isolated from the rest of migrant workers living in the dormitories, thus contact frequency remained relatively unchanged during the lockdown (circuit breaker) period.

**Social distancing with low isolation rate.** Under this scenario, social distancing interventions were implemented on day 72 after the first confirmed case of COVID-19 and lasted for two months after which contact frequency increased gradually over the simulation time. As a consequence, contact per person is assumed to decrease from 8 close contacts per persons per day to 4 and begins to increase gradually after the lockdown; the contact frequency for the migrant workers is assumed to remain unchanged at 10 close contact per person per day. In addition, it was assumed that 20% of diagnosed asymptomatic and symptomatic individuals under the Singapore containment approach will be diagnosed and isolated under this scenario. The difference between the Singapore containment approach and the social distancing with low isolation rate is that, under the social distancing with low isolation rate, only 20% of diagnosed asymptomatic and symptomatic individuals are quarantined, while that for the Singapore containment approach is 100%.

**Social distancing with moderate isolation rate.** This counterfactual scenario is similar to the previous social distancing intervention, except that 40% of diagnosed asymptomatic and symptomatic individuals under the Singapore containment approach will be diagnosed and isolated under this scenario. The difference between the social distance with moderate isolation rate, the social distancing with low isolation rate and the Singapore containment approach are that, under the social distancing with moderate isolation rate, only 40% of diagnosed asymptomatic and symptomatic individuals are quarantined, while that for social distancing with low rate is 20% and Singapore containment approach is 100%.

## Model validation and sensitivity analysis

We compared our simulated new cases and cumulative confirmed COVID-19 cases in Singapore for the local population and migrant workers with official data from the Ministry of Health, Singapore (see Fig 4) from the onset of the epidemic (January 23, 2020) to June 7, 2020. For sensitivity analysis, multivariate sensitivity analysis that varies selected parameters by ±25% was performed with random draws from uniform distributions over the designated

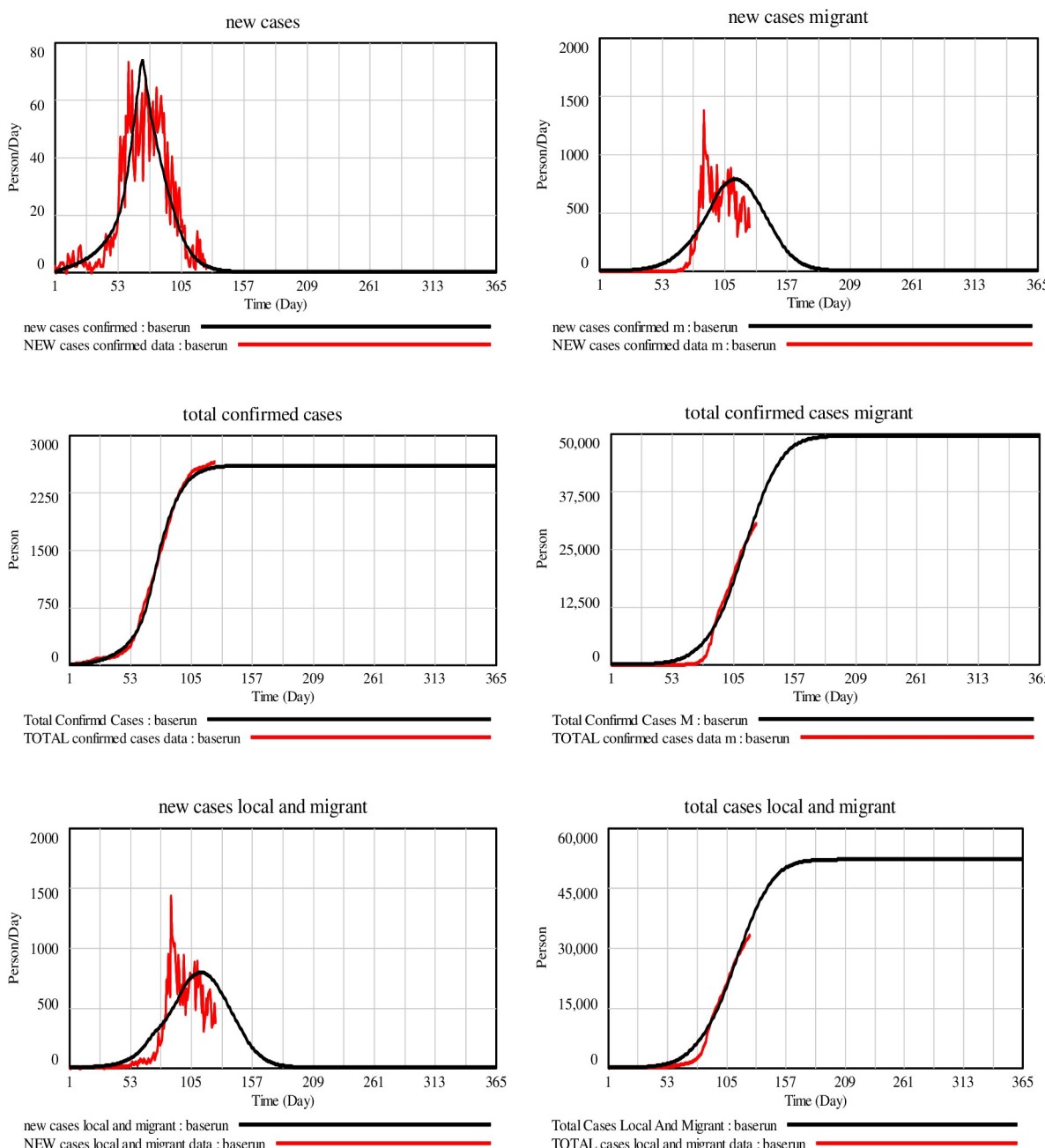

**Fig 4. Comparing simulated confirmed new cases and cumulative confirmed cases to data.**

range to explore how a change in these parameters influences the outcome variables namely cumulative infected cases, deaths, and fraction infection (see S5–S7 Appendices in S1 File).

## Results

The results of the simulation in presented in Table 2 and Fig 5. The reference case (Singapore Intervention), in which the model simulates containment interventions actually implemented in Singapore, closely tracked historical trends of COVID-19 infection from January 23, up to

**Table 2. Projected time to peak infection, duration of infection, cumulative infection, proportion infected and total deaths.**

| Interventions | Cumulative infected Cases (person) | | % of population infected | | Total deaths (person) |
|---|---|---|---|---|---|
| | Confirmed | Actual | Confirmed | Actual | |
| Singapore Intervention | 52,053 [49,370–54,735] | 86,041 [81,097–90,986] | 0.87% [0.83%-0.92%] | 1.45% [1.36%-1.53%] | 37 [34–39] |
| Mitigation with Low Isolation Rate | 67,539 [64,245–70,832] | 260,420 [249,985–270,855] | 1.14% [1.08%-1.19%] | 4.38% [4.20%-4.55%] | 137 [129–145] |
| Mitigation with Moderate Isolation Rate | 65,266 [62,122–68,409] | 179,104 [170,065–188,143] | 1.10% [1.04%-1.15%] | 3.01% [2.86%-3.16%] | 90 [85–95] |

June 7, 2020 (see Fig 5). Under the Singapore Intervention, by the end of the current epidemic cycle without considering the possibility of a second wave, the confirmed number of Singaporeans infected with COVID-19 (diagnosed asymptomatic and symptomatic cases) is projected to be 52,053 (with 95% confidence range of 49,370–54,735) representing 0.87% (0.83%-0.92%) of the total population; the actual number of Singaporeans infected with COVID-19 (diagnosed and undiagnosed asymptomatic and symptomatic infected cases) is projected to be 86,041 (81,097–90,986), which is 1.65 times the confirmed cases and represents 1.45% (1.36%-1.53%) of the total population. A peak in infected cases is projected to have occurred on around day 125 (27/05/2020) for the confirmed infected cases and around day 115 (17/05/2020) for the actual infected cases. The number of deaths is estimated to be 37 (34–39) among those infected with COVID-19 by the end of the epidemic cycle; consequently, the perceived case fatality rate is projected to be 0.07%, while the actual case fatality rate is estimated to be 0.043%.

In comparison, a mitigation intervention with 20% isolation rate was projected to increase the confirmed infected cases by 29.7% (23.4%-36.0%), increase the actual infected cases by 202% (190.5%-214.7%), move the time to peak infection for confirmed infected cases by 27 days earlier (30/04/2020) and that for the actual infected cases by 28 days earlier (19/04/2020) compared to the Singapore Intervention. In addition, the number of Singaporeans estimated to die from COVID-19 will increase 3.7 (3.4–3.9) fold relative to the Singapore Intervention by the end of the epidemic. As a result, under this intervention, the perceived case fatality rate is estimated to be 0.20%, while the actual case fatality rate is projected to be 0.05%. Under a mitigation intervention with 20% isolation rate, the actual infected cases are projected to be 3.0 (2.9–3.1) fold relative to what it would have been under the Singapore intervention.

Likewise, a mitigation intervention with 40% quarantine rate, compared to the Singapore specific containment intervention could increase the confirmed cases by 25.3% (19.3%-31.4%), increase actual cases by 108% (97.6%-118.6%), move the time to peak infection by 19 days earlier (9/05/2020 for confirmed cases and 29/04/2020 for actual infected cases) for both the confirmed and actual infected cases, and increase the number of deaths 2.45-folds. As a result, the perceived case fatality rate is estimated to be 0.13% whereas the actual case fatality rate is estimated to be 0.05%. Equally, the actual infected cases are projected to be 2.0 (1.9–2.2) fold of the cases estimated under the Singapore intervention.

In addition to the counterfactual analysis, we explored the impact of: (a) immunity on the scenarios explored (see S8 Appendix in S1 File for how the immunity assumptions were implemented in the model and the simulation results), and (b) what would have happened if Singapore allowed the virus to take its natural course without intervention (i.e., uninhibited spread/ herd immunity) approach (see S9 Appendix in S1 File for how the uninhibited spread assumptions were implemented in the model and the simulation results). The results from our simulation model when the possibility of reinfection (using 6 months' immunity duration) is implemented suggest that the projected numbers remain relatively unchanged and a likelihood of second wave under the mitigation with low isolation intervention at the end of the

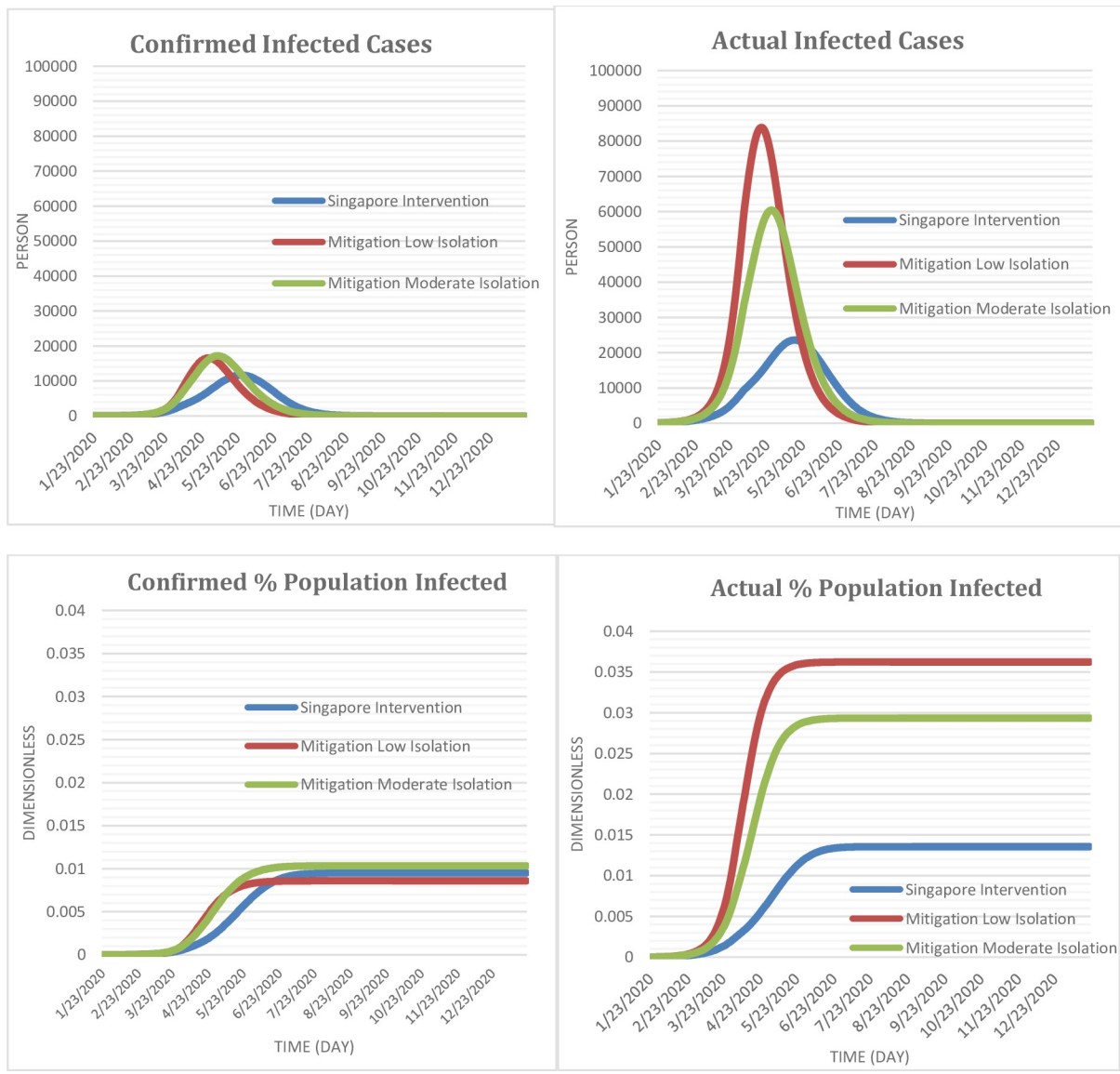

**Fig 5. Projected actual and confirmed cases of COVID-19; as well as the projected actual and confirmed percentage of the population infected in Singapore under current containment intervention and alternative mitigation interventions.**

simulation time; as expected, the only difference is that the recovered population decreases as individuals' transition from recovered to susceptible health state. This result supports our assumption that reinfection will have limited impact within the context of our modelling study. Further modelling exploration shows that had Singapore implemented a strategy of no active intervention (uninhibited spread leading to herd immunity), the number of Singaporeans infected with COVID-19 is projected to be 4.69 (4.64–4.74) million, representing 78.84% (77.99%-79.70%) of the total population. A peak in infected cases is projected to occur around day 154 (25/06/2020) and the number of deaths is estimated to be 2,466 (2,385–2,547) among those infected with COVID-19 by the end of the epidemic cycle assuming the current death rate remains unchanged.

## Discussion

In this study, we developed a COVID-19 infection model to explore what the trajectory of COVID-19 infection might have been in Singapore had the government intervention not focused on containment, but rather on mitigation. Compared to projections of a model calibrated to actual Singapore data based on a prompt and aggressive containment strategy, the simulation results indicate that a mitigation approach would have resulted in early peak infection, and increased both the cumulative confirmed and actual infection cases and deaths. Importantly, our simulation model result suggests that there about 65% more COVID-19 infection cases in Singapore that have not been captured in the official reported numbers which could possibly be uncovered via a serological study. In addition, further modelling exploration suggests that: (a) assuming a possibility of reinfection in individuals will have limited impact on the simulation results; (b) a strategy that focuses on uninhibited spread of COVID-19 would delay the peak of infection and increase both the cumulative cases and deaths by orders of magnitude.

In this counterfactual modelling exercise, we found that what seems to work to significantly decrease infected COVID-19 cases is the early implementation of containment interventions that focuses on meticulous and swift contact tracing and individual-level quarantine, in addition to standard health advice on hand washing, wearing of face mask, and social distancing. What our model suggests is that implementation of social distancing without contact tracing and individual-level quarantine does not work well. The policy implication based on insight from our simulation model is that general public health measures have to be applied together with targeted, aggressive and rapid containment in order to expect to substantially reduce the number of people infected with COVID-19 and consequent mortality, and should be the preferred intervention for managing COVID-19 and future epidemic outbreaks.

It is important to note that Singapore's implementation of contact tracing and quarantine to stop the spread of the virus has not been easy. Given that some individual transmission may occur before development of significant symptoms, the Singapore quarantine policy expended substantial effort to identify all exposed individuals deemed to have close contact with a confirmed infected individual, not only symptomatic individuals. In addition, contract tracing has to be swift to reduce the delay time from onset to isolation. It is important to note that Singapore was able to show early success in containment.

However, a recent outbreak in crowded foreign worker dormitories in Singapore has rapidly escalated the number of cases. Massive efforts are currently ongoing to isolate, test, sort, re-house and treat patients on-site at these dormitories. Most cases are being managed at community isolation facilities. As this population is relatively young with little co-morbidity, it is expected that the actual numbers of cases needing intensive care will be low and mortality also correspondingly low. This recent turn of events suggests that due to the ability of COVID-19 to transmit in pre-symptomatic or even asymptomatic individuals, contact tracing and quarantine also has limitations and requires application combined with more general social distancing measures.

The simulation model used for this study has several limitations. First, the epidemiology of COVID-19 is still not fully understood in terms of transmission and infectivity of the virus. Thus, we had to calibrate important parameters such as duration of infectiousness for asymptomatic individuals. Second, to reduce the complexity of the model, migration dynamics of the Singapore population were not included in the model, though migration plays an important role in the spread of COVID-19. We note that individuals traveling into Singapore can be easily targeted for containment in comparison with larger countries with less easily controlled borders. Lastly, contact frequency and pattern are highly dynamic across different segments of

the population; however, an average contact frequency was used in the model to represent all individuals.

In addition, modelling studies are needed to examine the impact of health systems response to COVID-19 on vulnerable non-COVID-19 patients; this will allow us to better balance of the needs of the entire population in response to future outbreaks.

## Conclusion

This study demonstrates that contact tracing, testing and aggressive containment has likely been the key to suppressing the number of COVID-19 infections in Singapore. These interventions should be combined with social distancing in the intervention packages currently being implemented across all countries and in future epidemic. Social distancing, though vital in slowing the growth of COVID-19, will be much less effective alone unless complemented with aggressive containment.

## Supporting information

**S1 File.**
(DOCX)

**S1 Data.**
(XLSX)

**S2 Data.**
(XLSX)

**S1 Dataset.**
(MDL)

## Author Contributions

**Conceptualization:** John P. Ansah, David Bruce Matchar, Jenny G. Low, Marcus Eng Hock Ong.

**Data curation:** Sean Lam Shao Wei, Jenny G. Low, Ahmad Reza Pourghaderi, Aloysius Chia Wei-Yan, Marcus Eng Hock Ong.

**Formal analysis:** John P. Ansah.

**Investigation:** Jenny G. Low.

**Project administration:** Sean Lam Shao Wei, Fahad Javaid Siddiqui, Tessa Lui Shi Min.

**Resources:** Ahmad Reza Pourghaderi.

**Validation:** John P. Ansah, Aloysius Chia Wei-Yan.

**Visualization:** Aloysius Chia Wei-Yan.

**Writing – original draft:** John P. Ansah, Tessa Lui Shi Min.

**Writing – review & editing:** John P. Ansah, David Bruce Matchar, Fahad Javaid Siddiqui, Tessa Lui Shi Min, Marcus Eng Hock Ong.

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
