## [Decision Letter · Decision Letter 0]

17 Nov 2020

PONE-D-20-19492

The Effectiveness of Public Health Interventions Against COVID-19: Lessons from Singapore Experience.

PLOS ONE

Dear Dr. Ansah,

Thank you for submitting your manuscript to PLOS ONE. After careful consideration, we feel that it has merit but does not fully meet PLOS ONE’s publication criteria as it currently stands. Therefore, we invite you to submit a revised version of the manuscript that addresses the points raised during the review process.

We look forward to receiving your revised manuscript.

Kind regards,

Kannan Navaneetham, PhD

Academic Editor

PLOS ONE

Journal Requirements:

2. In your ethics statement in the Methods section and in the online submission form, please clarify whether all data were fully anonymized before you accessed them.

Reviewers' comments:

Reviewer's Responses to Questions

**Comments to the Author**

1. Is the manuscript technically sound, and do the data support the conclusions?

Reviewer #1: Yes

Reviewer #2: Partly

2. Has the statistical analysis been performed appropriately and rigorously? 

Reviewer #1: Yes

Reviewer #2: I Don't Know

3. Have the authors made all data underlying the findings in their manuscript fully available?

Reviewer #1: Yes

Reviewer #2: No

4. Is the manuscript presented in an intelligible fashion and written in standard English?

Reviewer #1: Yes

Reviewer #2: Yes

5. Review Comments to the Author

Reviewer #1: The authors study the effectiveness of different interventions in the context of COVID-19 spread in Singapore. They distinguish between containment and mitigation, and use a modified SIR model to show the difference in total infections, peak load and number of deaths. One of the main contributions of the study is also an estimate of total COVID-19 infections (diagnosed and undiagnosed).

In addition to modeling contributions, the article also summarizes the early COVID-19 response in Singapore (e.g., Figure 1), and is a useful historical record. Though dated, the work in this paper is quite useful in highlighting the importance of timely and appropriate response to the pandemic.

Clarifications/Suggestions:

- One of the modifications to SIR model involves differentiating between the detected and undetected individuals. They also account for different infection stages. In terms of description, stating that 'symptomatic infected individuals were disaggregated into two infection stages' seems ambiguous. I think it is cleaner to state that the latent stage is shared by asymptomatic and symptomatic branches.

- Since a lot of the article hinges on the distinction between containment and mitigation, it would be good to add citations to align with the literature. For instance (https://jamanetwork.com/journals/jama/fullarticle/2764956) and (https://jamanetwork.com/journals/jama/fullarticle/2763187) are useful to mention.

- The authors report projected cases by end of current epidemic cycle, however no specific date is mentioned. Likewise, the peak timings are reported in terms of simulation days, than calendar time. It would be helpful if they could use actual months/dates.

- perceived and actual case fatality rate -> case fatality rate and infection fatality rate?

- In Background of Abstract, should it be 'we estimate the actual COVID-19 infections', given that confirmed case counts are publicly available.

- pg. 7 line 82 - Should be Figures 2 and 3.

- It would be good to provide an equation for \\beta given it implicitly captures the infectious population (is the proportion of total population?).

- The authors state the models were validated using case data until June 7th 2020. Would be useful to show these curves in Figure 5, along with the simulated curves (these are in S3, but useful in main paper to show the model fit).

- Though there are multiple parameters listed for the undiagnosed to diagnosed transition, none of them capture the detection probability. Is the fraction of detected infections just a function of the transition rates, or are there any assumptions about the testing protocol?

- How are the uncertainty bounds for the results obtained? Are these same as the +/-25% for selected parameters (the bounds in S8 seem to be much higher)? Which parameters were varied this way?

- The authors state that uninhibited spread would have lead to ~80% of the population infected. This is much higher than the compared scenarios, where the highest is around 4.5%. Is this based on R0=2 and remaining parameters unchanged? What about some of the diagnosis and isolation parameters?

Reviewer #2: 

General comment

The paper "The Effectiveness of Public Health Interventions Against COVID-19: Lessons from Singapore Experience" aims at carring out a counterfactual analysis on the spread of COVID-19 infections in Singapore under different scenarios.

The paper is interesting, well organised and well written for the most part, however several methodological details should be produced and clarified in order to make the anayisis transparent and reproducible, and ultimately, make the reader able to judge on its validity. To this end, data and source code that generated estimates and simulations should be shared with reviewers at least. In the rest of this document, I enumerate the main issues that should be tackled. In the last section, some typos are reported, however a general revision of the manuscript is needed.

In conclusion, I think that the paper is worth of publication in PLOS ONE once the following major questions and issues are addressed.

Major issues

1. [sect. "Methods"] It should be clearly stated what amendments authors made to the SIR or the SIDARTHE model, they should be clearly motivated, and their implications discussed.

2. [pp. 8-9] All equations should be carefully revised. Unlike reference models such as SIR or SIDARTHE, and unlike what stated in the paper (line 111), none of equations (1)-(14) is a differential equation. Moreover, I suggest authors to revise the mathematical notation by including only latin and greek characters: this would improve the readability of the paper. In any case, superscript-like characters such as the superscript turned m and the superscript delta (which are phonetic symbols) should be definitely avoided, as they may be confused with exponents.

3. The meaning and the role of parameters yat (Ѣ) and phi (Φ) in the model is not clear to me. For example, why only a portion yat of diagnosed people is quarantined? Moreover, parameter phi is meant to be "onset to isolation delay" throughout all the paper (except on page 9 where it is defined, and the definitions of phi and yet are switched), however it is not clear to me what term "onset" refers to, expecially if I examine the diagram in Figure 3.

4. [p. 11] It should be clearly explained how "endogenous" parameters are determined, and how calibration is performed.

5. [sect. "Simulated Interventions"] Differences and shared aspects of the scenarios should emerge more clearly, so as to make them more comparable. If choices on the parameters and conditions which distinguish each scenario are justified (when possible), this would definitely improve the validity of the analysis.

6. Authors state that they performed sensitivity analysis, however no comments to the results are provided neither in the manuscript not in the supplementary material.

7. It is not clear how confidence intervals in section "Results" are computed, as the model authors describe is deterministic.

8. On page 14, authors claim that they “explored the impact of (a) immunity on the scenarios explored, and (b) what would have happened if Singapore allowed the virus to dake it natural course without intevention approach”, however no results are provided either in the manuscript of in the supplementary material. Moreover they do not explain how they modified the model (and set the parameters) in order to relax the immunisation hypothesis.

Minor issues

1. Figures S1 and S2 of the Supplementary Material are missing. All the supplementary file should be revised, as there are several figures which are not completely readable, wherease captions and title graphs do not adequately clarify what information is plotted.

2. Second and third items on page 9 would be more easy-to-read if parameters were reported (also ?) along with definitions.

Some typos and other minor details

1. [p. 3, sect. "Why this study was done?", point 1, 3rd line] A quotation mark is missing for keyword "2019 nCoV".

2. [p. 3, sect. "Why this study was done?", point 3] Last sentence says "The list of 30 papers included are provided in the appendix". If authors refer to Appendix S10 of the supplementary material, they could explicitly cite the reference label "S10". Moreover, Appendix S10 actually consists of 31 instead of 30 references.

3. [line 8] Please, specify the date adverb "currently" refers to.

4. [line 82] I think that figures authors refer to are those with numbers 2 and 3.

5. [page 8] Equation number (11) has not been used.

6. [equation 13] Right parenthesis is missing.

7. [line 146] Remove comma after "alpha".

8. [line 154] "denotes" should be "denote", and a space is missing before "kappa".

9. [line 163] "deMotes" should be "denote".

10. [line 213] "will diagnosed" should be "will be diagnosed".

11. [line 220] "...these parameters influence..." should be "...these parameters influences...".

12. [line 260] "it natural course" should be "its natural course".

6. PLOS authors have the option to publish the peer review history of their article (what does this mean?). If published, this will include your full peer review and any attached files.

Reviewer #1: No

Reviewer #2: No

---

## [Author Response · Author response to Decision Letter 0]

16 Feb 2021

Response to Reviewers

REVIEWER #1

The authors study the effectiveness of different interventions in the context of COVID-19 spread in Singapore. They distinguish between containment and mitigation, and use a modified SIR model to show the difference in total infections, peak load and number of deaths. One of the main contributions of the study is also an estimate of total COVID-19 infections (diagnosed and undiagnosed).

In addition to modelling contributions, the article also summarizes the early COVID-19 response in Singapore (e.g., Figure 1), and is a useful historical record. Though dated, the work in this paper is quite useful in highlighting the importance of timely and appropriate response to the pandemic.

Clarifications/Suggestions:

Comment #1: One of the modifications to SIR model involves differentiating between the detected and undetected individuals. They also account for different infection stages. In terms of description, stating that 'symptomatic infected individuals were disaggregated into two infection stages' seems ambiguous. I think it is cleaner to state that the latent stage is shared by asymptomatic and symptomatic branches.

Response: Thanks for the suggestion. We appreciate and understand your suggestion, however, upon critical examination, we felt that our current description will improve understanding of the mechanics of the model given the different stages for each health state. Hence, we decided to keep the description as it is. 

Comments #2: Since a lot of the article hinges on the distinction between containment and mitigation, it would be good to add citations to align with the literature. For instance (https://jamanetwork.com/journals/jama/fullarticle/2764956) and (https://jamanetwork.com/journals/jama/fullarticle/2763187) are useful to mention.

Response: Thanks for the suggestion. We have included the citations in the list of references and simulated interventions section of the manuscript. 

Comments #3: The authors report projected cases by end of current epidemic cycle, however no specific date is mentioned. Likewise, the peak timings are reported in terms of simulation days, than calendar time. It would be helpful if they could use actual months/dates.

Response: Thanks for the suggestion. We have included calendar time to the reported peak times. 

Comments #4: Perceived and actual case fatality rate -> case fatality rate and infection fatality rate?

Response: Thanks for the comment. The perceived case fatality rate is the number of deaths divided by confirmed cases; while the actual case fatality is the number of deaths divided by the actual cases (confirmed and unconfirmed infections). 

Comments #5: In Background of Abstract, should it be 'we estimate the actual COVID-19 infections', given that confirmed case counts are publicly available.

Response: Thanks for the suggestion. We have made the changes as suggested as follows:

“In addition, we estimate the actual COVID-19 infection cases in Singapore, given that confirmed cases are publicly available.”

Comments #6: pg. 7 line 82 - Should be Figures 2 and 3.

Response: Thanks for the correction. We have made the changes to figure 2 and 3.

Comments #7: It would be good to provide an equation for \\beta given it implicitly captures the infectious population (is the proportion of total population?).

Response: Thanks for the suggestion. The equation for beta (β which is now "b") is “infected asymptomatic undiagnosed infectious” and “infected symptomatic undiagnosed infectious”, divided by “total population”.

Comments #8: The authors state the models were validated using case data until June 7th 2020. Would be useful to show these curves in Figure 5, along with the simulated curves (these are in S3, but useful in main paper to show the model fit).

Response: Thanks for the suggestion. We have moved the validation graphs to the main paper as suggested. 

Comments #9: Though there are multiple parameters listed for the undiagnosed to diagnosed transition, none of them capture the detection probability. Is the fraction of detected infections just a function of the transition rates, or are there any assumptions about the testing protocol?

Response: Thanks for the comment. Yes, the detected infections are just a function of the transition rates—as infected individuals move from undiagnosed infected to diagnosed infected. Thus, the detection probability is reflected in the “onset to isolation delay” parameter since as clearly indicated in the manuscript, in Singapore, all confirmed COVID-19 cases are isolated. 

Comments #10: How are the uncertainty bounds for the results obtained? Are these same as the +/-25% for selected parameters (the bounds in S8 seem to be much higher)? Which parameters were varied this way?

Response: Thanks for the comment. On the parameters included in the sensitivity analysis, table 1 clearly indicates the parameters included in the sensitivity analysis with (*). These parameters are:

 Probability of infection given contact 

 Latent period

 Duration of infectiousness asymptomatic 

 Onset to isolation delay

 Incubation time

 Recovery time diagnosed asymptomatic 

 Recovery time undiagnosed asymptomatic

 Duration of infectiousness symptomatic

 Symptomatic development to care delay 

 Mortality rate undiagnosed symptomatic

 Mortality rate diagnosed symptomatic

 Recovery time undiagnosed symptomatic

 Recovery time diagnosed symptomatic

 Fraction asymptomatic without symptoms

The uncertainty bounds were obtained as follows: 

 First, the sensitivity analysis was performed by varying the list of parameters included in the sensitivity analysis by ±25%; and the simulation model was run 500 times.

 Second, the results were exported to excel and basic statistical analysis was performed to obtain 95% confidence interval around the mean values. The bounds in S7, S8 and S9 seems much higher because it includes the outliers. The reported uncertainty bounds in table 2 is the 95% confidence interval from the statistical analysis of the sensitivity analysis. 

Comments #11: The authors state that uninhibited spread would have led to ~80% of the population infected. This is much higher than the compared scenarios, where the highest is around 4.5%. Is this based on R0=2 and remaining parameters unchanged? What about some of the diagnosis and isolation parameters?

Response: Thanks for the comment. Under the uninhibited spread strategy, diagnosis and isolation is assumed to be non-existent; and where diagnosis is assumed to be present, individuals diagnosed with COVID-19 are not isolated and R0=2. Thus, it is assumed that isolation of infected individuals is assumed to be absent. 

REVIEWER #2

The paper "The Effectiveness of Public Health Interventions Against COVID-19: Lessons from Singapore Experience" aims at carrying out a counterfactual analysis on the spread of COVID-19 infections in Singapore under different scenarios.

The paper is interesting, well organised and well written for the most part, however several methodological details should be produced and clarified in order to make the analysis transparent and reproducible, and ultimately, make the reader able to judge on its validity. To this end, data and source code that generated estimates and simulations should be shared with reviewers at least. In the rest of this document, I enumerate the main issues that should be tackled. In the last section, some typos are reported, however a general revision of the manuscript is needed.

In conclusion, I think that the paper is worth of publication in PLOS ONE once the following major questions and issues are addressed.

Major issues

Comments #1: [sect. "Methods"] It should be clearly stated what amendments authors made to the SIR or the SIDARTHE model, they should be clearly motivated, and their implications discussed.

Response: Thanks for the comment. We have added the following statement in the methods section to address the difference between the modified SIR model used and the SIDARTHE model. 

“The main differences between the proposed modified SIR model presented herein and the SIDARTHE model are that: (a) our model has 7 health states compared to the eight-health state of the SIDARTHE model. (b) our modified SIR model further disaggregates the asymptomatic infected individuals into three infection stages, which is not part of the SIDARTHE model. (c) the modified SIR model disaggregates the symptomatic infected individuals into two infection stages, which is absent in the SIDARTHE model. We strongly believe that the amended made compared to the SIDARTHE model will significantly improve the accuracy of the simulation model to predict the infection trajectory of COVID-19, and more importantly, help to estimate the actual COVID-19 infection cases.”

Comments #2: [pp. 8-9] All equations should be carefully revised. Unlike reference models such as SIR or SIDARTHE, and unlike what stated in the paper (line 111), none of equations (1)-(14) is a differential equation. Moreover, I suggest authors to revise the mathematical notation by including only Latin and Greek characters: this would improve the readability of the paper. In any case, superscript-like characters such as the superscript turned m and the superscript delta (which are phonetic symbols) should be definitely avoided, as they may be confused with exponents.

Response: Thanks for the comments. We have changed the equations to differential equations and also changed notations as suggested by the reviewer. 

Comments #3: The meaning and the role of parameters yat (Ѣ) and phi (Φ) in the model is not clear to me. For example, why only a portion yat of diagnosed people is quarantined? Moreover, parameter phi is meant to be "onset to isolation delay" throughout all the paper (except on page 9 where it is defined, and the definitions of phi and yet are switched), however it is not clear to me what term "onset" refers to, especially if I examine the diagram in Figure 3.

Response: Thanks for the comment. yat (Ѣ, which is now “p”) is the fraction quarantined and is defined as “total confirmed cases” divided by “total infected cases”. This parameter yat (Ѣ, which is now “p”) estimates the fraction of total infected cases confirmed, this is important because there is a delay between getting infected and being diagnosed (confirmed) which is captured by the onset to isolation delay phi (Φ, which is now “x”). Thus, phi (Φ, which is “x”) is onset to isolation delay, which is the time it takes for someone infected with COVID-19 to be diagnosed and isolated. As indicated in reference [21], this parameter decreased from 9 days early in the outbreak to 2 days. 

On page 9, the definition for yat (Ѣ, which is now “p”) and phi (Φ, which is now “x”) was not switched. The definitions are consistent with what we have in the manuscript. On page 9, yat (Ѣ, which is now “p”) is fraction quarantined and phi (Φ, which is now “x”) is onset to isolation delay. 

Comments #4: [p. 11] It should be clearly explained how "endogenous" parameters are determined, and how calibration is performed.

Response: Thanks for the comment. We have added the definition of the endogenous parameters in the manuscript under the model input section. The definitions are: 

Proportion of undiagnosed infected (b) = (Infected symptomatic undiagnosed infectious + Infected 

 Asymptomatic undiagnosed infectious)/Total population

Fraction quarantined (p) = Total confirmed cases / Total infected cases 

Comments #5: [sect. "Simulated Interventions"] Differences and shared aspects of the scenarios should emerge more clearly, so as to make them more comparable. If choices on the parameters and conditions which distinguish each scenario are justified (when possible), this would definitely improve the validity of the analysis.

Response: Thanks for the comments, we have added few sentences, to clarify the differences between the simulated interventions. 

Comments #6: Authors state that they performed sensitivity analysis, however no comments to the results are provided neither in the manuscript not in the supplementary material.

Response: Thanks for the comment. Table 2 shows the results with confidence intervals. The confidence intervals were obtained via the sensitivity analysis. 

Comments #7: It is not clear how confidence intervals in section "Results" are computed, as the model authors describe is deterministic.

Response: Thanks for the comment. The confidence intervals were estimated from the sensitivity analysis output. As explained in the manuscript, multivariate sensitivity analysis that varies selected parameters by ±25% (in Table 1, parameters with * were included in the sensitivity analysis) was performed with random draws from uniform distributions over the designated range to explore how a change in these parameters influences the outcome variables namely cumulative infected cases, deaths, and fraction infection. The simulation model was run 500 times and the output was exported to excel for analysis. The mean, and 95% confidence interval were estimated and reported in Table 2. 

Comments #8: On page 14, authors claim that they “explored the impact of (a) immunity on the scenarios explored, and (b) what would have happened if Singapore allowed the virus to take it natural course without intervention approach”, however no results are provided either in the manuscript of in the supplementary material. Moreover, they do not explain how they modified the model (and set the parameters) in order to relax the immunisation hypothesis.

Response: Thanks for the comment. We have provided the results and model changes made for the immunity assumption and the uninhibited spread/herd immunity assumptions. S8 provides details for the immunity assumption and S9 for the uninhibited spread/herd immunity assumption. 

Minor issues

Comments #9: Figures S1 and S2 of the Supplementary Material are missing. All the supplementary file should be revised, as there are several figures which are not completely readable, whereas captions and title graphs do not adequately clarify what information is plotted.

Response: Thanks for the suggestion. We have reorganised the supplementary file. 

Comments #10: Second and third items on page 9 would be more easy-to-read if parameters were reported (also ?) along with definitions.

Response: Thanks for the suggestion. Because table 1 list all these parameters with their values, we are of the opinion we do not need to provide it again in the manuscript. 

Some typos and other minor details

Comments #10: [p. 3, sect. "Why this study was done?", point 1, 3rd line] A quotation mark is missing for keyword "2019 nCoV".

Response: Thanks for the suggestion. We have made the correction. 

Comments #10: [p. 3, sect. "Why this study was done?", point 3] Last sentence says "The list of 30 papers included are provided in the appendix". If authors refer to Appendix S10 of the supplementary material, they could explicitly cite the reference label "S10". Moreover, Appendix S10 actually consists of 31 instead of 30 references.

Response: Thanks for the suggestion. We have made the correction. 

Comments #11: [line 8] Please, specify the date adverb "currently" refers to.

Response: Thanks for the suggestion. We have made the correction. 

Comments #12: [line 82] I think that figures authors refer to are those with numbers 2 and 3.

Response: Thanks for the suggestion. We have made the correction. 

Comments #13: [page 8] Equation number (11) has not been used.

Response: Thanks for the suggestion. We have made the correction. 

Comments #14: [equation 13] Right parenthesis is missing.

Response: Thanks for the suggestion. We have made the correction. 

Comments #15: [line 146] Remove comma after "alpha".

Response: Thanks for the suggestion. We have made the correction. 

Comments #16: [line 154] "denotes" should be "denote", and a space is missing before "kappa".

Response: Thanks for the suggestion. We have made the correction. 

Comments #17: [line 163] "deMotes" should be "denote".

Response: Thanks for the suggestion. We have made the correction. 

Comments #18: [line 213] "will diagnosed" should be "will be diagnosed".

Response: Thanks for the suggestion. We have made the correction. 

Comments #19: [line 220] "...these parameters influence..." should be "...these parameters influence...".

Response: Thanks for the suggestion. We have made the correction. 

Comments #20: [line 260] "it natural course" should be "its natural course".

Response: Thanks for the suggestion. We have made the correction

---

## [Decision Letter · Decision Letter 1]

5 Mar 2021

The Effectiveness of Public Health Interventions Against COVID-19: Lessons from the Singapore Experience.

PONE-D-20-19492R1

Dear Dr. Ansah,

We’re pleased to inform you that your manuscript has been judged scientifically suitable for publication and will be formally accepted for publication once it meets all outstanding technical requirements.

Kind regards,

Kannan Navaneetham, PhD

Academic Editor

PLOS ONE

Additional Editor Comments (optional):

Reviewers' comments:

Reviewer's Responses to Questions

**Comments to the Author**

1. If the authors have adequately addressed your comments raised in a previous round of review and you feel that this manuscript is now acceptable for publication, you may indicate that here to bypass the “Comments to the Author” section, enter your conflict of interest statement in the “Confidential to Editor” section, and submit your "Accept" recommendation.

Reviewer #2: All comments have been addressed

2. Is the manuscript technically sound, and do the data support the conclusions?

Reviewer #2: Yes

3. Has the statistical analysis been performed appropriately and rigorously? 

Reviewer #2: Yes

4. Have the authors made all data underlying the findings in their manuscript fully available?

Reviewer #2: Yes

5. Is the manuscript presented in an intelligible fashion and written in standard English?

Reviewer #2: Yes

6. Review Comments to the Author

Reviewer #2: I am satisfied with the authors' answers and revision of the manuscript, which may be accepted for publication.

7. PLOS authors have the option to publish the peer review history of their article (what does this mean?). If published, this will include your full peer review and any attached files.

Reviewer #2: No

---

## [Editor Report · Acceptance letter]

15 Mar 2021

PONE-D-20-19492R1 

The Effectiveness of Public Health Interventions Against COVID-19: Lessons from the Singapore Experience 

Dear Dr. Ansah:

I'm pleased to inform you that your manuscript has been deemed suitable for publication in PLOS ONE. Congratulations! Your manuscript is now with our production department. 

Kind regards, 

on behalf of

Professor Kannan Navaneetham 

Academic Editor

PLOS ONE